# Anomalous Geomagnetic Signal Emphasized before the Mw8.2 Coastal Alaska Earthquake, Occurred on 29 July 2021

**DOI:** 10.3390/e24020274

**Published:** 2022-02-14

**Authors:** Dragoș Armand Stănică

**Affiliations:** Department of Electromagnetism and Lithosphere Dynamics, Institute of Geodynamic of the Romanian Academy, R-020032 Bucharest, Romania; armand@geodin.ro; Tel.: +40-788-419-041

**Keywords:** geomagnetic polarization parameters, precursory geomagnetic signals, Mw8.2 earthquake, CMO and NEW observatories—Alaska

## Abstract

A very strong earthquake of magnitude Mw8.2 struck the coastal zone of Alaska (USA), on 29 July 2021. This earthquake was felt around the Gulf of Alaska, on a wide offshore area belonging to USA and Canada. In order to identify an anomalous geomagnetic signal before the onset of this earthquake, we retrospectively analyze the data collected on the interval 17 June–31 July 2021, via internet, at the two geomagnetic observatories, College (CMO)—Alaska and Newport (NEW)—USA, by using the polarization parameter (BPOL) and the strain effect related to geomagnetic signal identification. Thus, for both observation sites (CMO and NEW), the daily mean distribution of the BPOL and its standard deviation (STDEV) are carried out using an FFT band-pass filtering in the ULF range (0.001–0.0083 Hz). Further on, a statistical analysis based on a standardized random variable equation is applied to emphasize the following: (a) the anomalous signature related to Mw8.2 earthquake on the both time series BPOL*(CMO) and BPOL*(NEW); (b) the differentiation of the transient local anomalies associated with Mw8.2 earthquake from the internal and external parts of the geomagnetic field, taking the NEW observatory as reference. Finally, on the BPOL*(NEW-CMO) time series, carried out on the interval 7–31 July 2021, a very clear anomaly of maximum, greater than 1.2 STDEV, was detected on 22 July, with 7 days before the onset of the Mw8.2 earthquake.

## 1. Introduction

Natural events such as earthquakes are inevitable and extremely hard to predict; therefore, the actual major challenge for the geoscience community is to develop specific methodologies that are able to identify reliable short-term pre-seismic anomalous signals, on the time scale of weeks, days or hours, which is believed to be the highest priority for social demands in seismic-active countries [1]. Consequently, in the last few decades, the electromagnetic/geomagnetic signals related to earthquakes have been identified based on ground-based and satellite observations techniques. The most known of them are: the European VLF/LF radio network [2]; plasma turbulence in the ionosphere prior to the earthquakes on the DEMETER registrations [3]; new magnetic index based on vertical geomagnetic variation [4]; ULF magnetic field measurement related to Mw7.1 Loma Prieta earthquake [5,6]; anomalous geomagnetic diurnal variation mainly in the z-component associated with the Mw9.0 Tohoku earthquake (which from Maxwell equations reflects the coexistence of seismic electrical activity [J. Geophys. Res. Space Physics 119, 9192–9206 (2014); Europhysics Letters (EPL) 132, 29001 (2020)]) [7]; multiple fractal model of pre-seismic electromagnetic phenomena [8]; ULF geomagnetic disturbances related to the earthquakes by using reference data [9]; ULF magnetic disturbances associated with earthquakes [10]; geophysical data possibly associated with the Ms8.0 Wenchuan earthquake [11]; electric and magnetic fields accompanying seismic and volcanic activity [12]; long-range anomalous electromagnetic effect related to Mw9.0 Great Tohoku earthquake, the epicenter of which could be estimated [Proc. Natl. Acad. Sci. USA 112, 986–989 (2015)] in advance by studying the spatiotemporal variations of the variability [Europhysics Letters (EPL) 91, 59001 (2010)] of the order parameter of seismicity in Japan [13]; earthquake prediction and precursor [14]; changes in the geoelectric conductivities, that in the real geophysical properties may generate internal electromagnetic effect [15,16,17,18,19,20]. All these mentioned research methods may supply more useful information about the origin of the different pre-seismic anomalous geomagnetic signals related to the major seismic events, what means a more conclusive separation of them. The aim of the present paper is to emphasize certain inter-relations between the geomagnetic anomalous geomagnetic signals and Mw8.2 earthquake occurrence on the coastal zone of Alaska—USA, on 29 July 2021, by using the geomagnetic data supplied by the College (CMO) and Newport (NEW) geomagnetic observatories. In order to identify the geomagnetic precursory anomalies, the BPOL(CMO), BPOL(NEW), ABS BPOL*(CMO), ABS BPOL*(NEW), and ABS BPOL*(NEW-CMO) were used.

## 2. Materials and Methods

### 2.1. Earthquake Location and Seismicity

In conformity with the data offered by Euro Mediterranean Seismic Centre, the map presented in Figure 1 reveals both the location of the Mw8.2 earthquake generated on the coastal zone of Alaska (USA) at about 43 km depth, on 29 July 2021 (red full circle), and the placement of the both geomagnetic observatories: College (CMO) and Newport (NEW) (green full circles). To have an idea about the circumstances of this earthquake occurrence, on the above-mentioned map the seismicity from the previous 7 days is presented in the area by means of yellow and brown circles. The present paper tries to show how this activity is reflected in the geomagnetic field variations by analyzing some geomagnetic monitoring data in the frame of the specific geological circumstances.

### 2.2. Basic Theoretical Concepts

Starting around the year 2010, geomagnetic monitoring investigations created the possibility to elaborate on a specific methodology capable of emphasizing certain interrelations between the pre-seismic ultralow frequency (ULF) anomalous geomagnetic signature and a major seismic event occurrence [13]. To tackle this methodology some theoretical concepts concerning earthquake generation mechanism had to be taken into consideration, such as: piezomagnetic effect, magneto-hydrodinamic effect and electrokinetic effect [18]. Thus, in order to identify any pre-seismic anomalous signature related to Mw.8.2 earthquake we focused on two relations:


(a)Polarization parameter (BPOL) expressed as:BPOL (f) = Bz (f)/SQRT[Bx^2^(f) + By^2^(f)](1)
where: Bx(f), By(f) and Bz(f) are horizontal and vertical components of the geomagnetic field;(b)Range effect of the strain related to the pre-seismic geomagnetic signals identification, due to the Mw8.2 earthquake, Morgunov and Malzev relation [8]:R (km) = 10 ^0.5 M−0.27^(2)
where: R is epicentral distance (R is the distance between the earthquake location and the CMO and NEW Geomagnetic Observatories) and M is the earthquake magnitude. According to relation (2) “the strain effect” due to the Mw8.2 earthquake was felt for R ≈ 6760 km. As both epicentral distances are less (R ≈ 1200 km for CMO observatory and R ≈ 2400 km for NEW observatory), the identification criteria for an anomalous pre- seismic signal is fulfilled.


### 2.3. Data Collection, Processing and Analysis

The geomagnetic data (Bx, By and Bz) obtained, via internet (www.intermagnet.org), from the geomagnetic observatories College (CMO) and Newport (NEW), on the interval 17 June–31 July 2021, are used:(a)To carry out the daily mean distribution of the Polarization Parameter (BPOL) and its Standard Deviation (STDEV)—see formula on the web, Relation (1);(b)To emphasize a possible pre-seismic geomagnetic signature related to the Mw8.2 earthquake, applying a statistical analysis based on relation (3)
BPOL* = (X − Y)/W(3)
where: X is the value of the BPOL starting with 1 July 2021 till 31 July 2021, Y is 30 days running average of BPOL for consecutive days before a specific day, W is the same as Y but for STDEV obtained for 30 consecutive days before the specific day, BPOL* emphasizes the threshold for anomaly using STDEV. The explanation for (X, Y, W) and BPOL* may be seen detailed in [19].(c)To separate the pre-seismic anomalous signals from ionospheric and terrestrial variations of the geomagnetic field, the Relation (4) was applied, where the Newport geomagnetic observatory was taken as reference
BPOL*(NEW-CMO) = (A − B)/C (4)
where: A represents the difference between BPOL NEW and BPOL CMO for a specific day, on the interval 7 July 2021–31 July 2021, B is 30 days running average of (BPOL NEW- BPOL CMO) before a specific day, C is 30 days running average of (STDEV NEW-STDEV CMO) before a specific day. BPOL*(NEW-CMO) time series emphasizes the threshold for anomaly using STDEV. For more explanations for A, B and C in (4), see Reference [19].

## 3. Results

After the strain effect was determined in conformity with Relation (2) and the identification criteria for an anomalous pre-seismic signal was fulfilled, geomagnetic monitoring data processing by using Fast Fourier Transform—Band Pass Filtering Analysis in the ultralow frequency range (0.001–0.0083 Hz) will supply the necessary information concerning the daily distribution of the polarization parameter BPOL for the two CMO and NEW observatories.

### 3.1. Time Series Distribution for BPOL(COM) and BPOL(NEW)

The daily mean distribution of the BPOL and its STDEV obtained for the College (CMO) and Newport (NEW) observatories, on the interval 1–31 July 2021, will supply information regarding the geomagnetic field variations and, most importantly, will emphasize if their values exceeded the standard limit. This situation may be seen in the Figure 2 and Figure 3, where the daily mean distribution of the BPOL and its STDEV, for the CMO observatory and Newport observatory, respectively, is presented in the time interval 1–31 July 2021.

### 3.2. Time Series Distribution for BPOL*(COM) and BPOL*(NEW)

The pre-seismic anomalous intervals associated to the Mw8.2 earthquake are revealed in the following time series: BPOL*(CMO), BPOL*(NEW) and BPOL*(NEW-CMO), in conformity with Figure 4, Figure 5 and Figure 6.

## 4. Conclusions

After the strain effect was determined in conformity with Relation (2) and the identification criteria for an anomalous pre-seismic signal was fulfilled, geomagnetic monitoring data processing by using Fast Fourier Transform—Band Pass Filtering analysis in the ultralow frequency range (0.001–0.0083 Hz) supplied the necessary information concerning the daily mean distribution of the polarization parameter BPOL for the two CMO and NEW observatories. To take into consideration all the information carried out on the base of the geomagnetic data collected from the College and Newport (Alaska—USA), the final results were very useful, emphasizing a precursory geomagnetic anomaly 7 days before the Mw8.2 earthquake occurrence, as follows:-The BPOL (CMO) and BPOL(NEW) time series distributions obtained on the interval 1–31 July 2021 (Figure 2 and Figure 3) highlight an obvious geomagnetic anomaly extended to a three-day interval, with an apex on 22 July, 7 days before the earthquake;-The same precursory signal may be also seen on ABS BPOL*(CMO) and ABS BPOL*(NEW) carried out on the interval 7–31 July, by means of the Relation (3), revealing an asimetric and more narrow geomagnetic anomaly of maxim, extended on the interval 18–24 July, with an apex on 22 July, 7 days before the Mw8.2 earthquake;-Information concerning the separation of the pre-seismic signal by the solar storm effect (Kp) was obtained with the Relation (4), where the Newport was taken as reference. The ABS BPOL*(NEW-CMO) time series distribution highlights an apex of the geomagnetic anomalous signal, with 1.3 magnitude on 22 July, identified by means of a threshold for anomaly in Figure 6 (red dashed line).

As final conclusion, it is obvious that the last time series ABS BPOL*(NEW-CMO) demonstrates that the applied methodology, based on the two geomagnetic monitoring sites, one of them taken as reference, might supply useful information to identify, with high accuracy, a pre-seismic anomalous signal with a value of 1.3 (Figure 6) on 22 July 2021, with 7 days before the strong Mw8.2 earthquake.

For a higher accuracy there is a neccessity to use more monitoring sites, so that many directions of the seismic waves propagation be covered, in conformity with specific tectonical features of the investigated area.

It is also worth mentioning that, generally, the lead time of the precursory geomagnetic signal on the time scale may be of weeks, days or hours, as a consequence of the tectonical structure and many complex (physico-chemical, mechanical and geodynamic) processes.

## Figures and Tables

**Figure 1 entropy-24-00274-f001:**
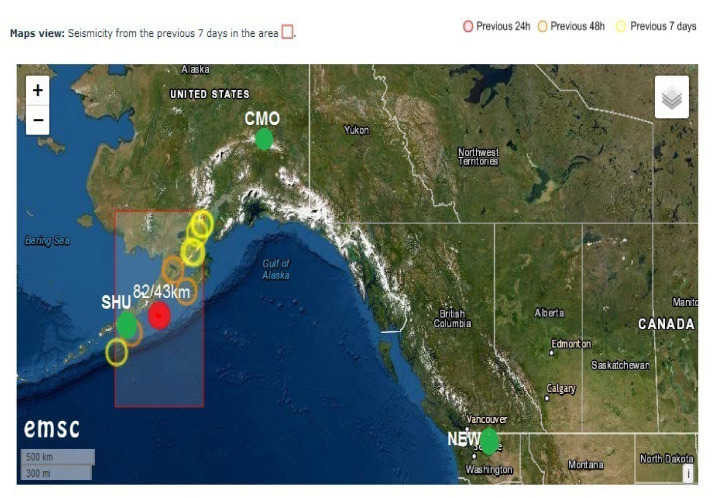
Mw8.2 earthquake location (red full circle) and the placement of the two observatories: CMO and NEW (green full circles).

**Figure 2 entropy-24-00274-f002:**
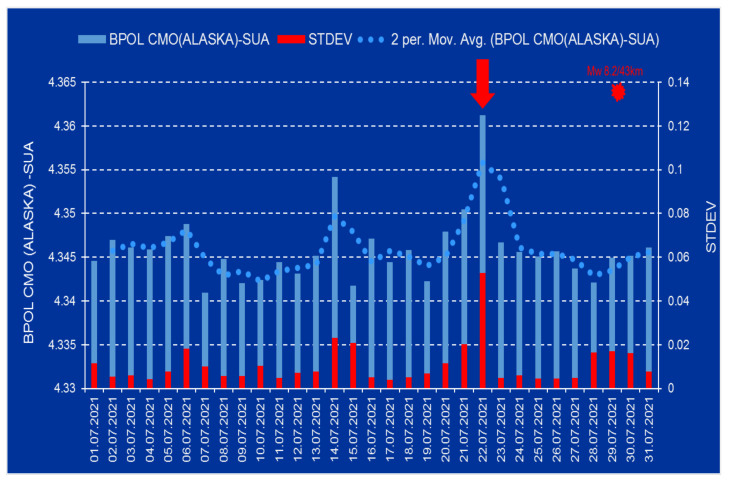
BPOL and STDEV time series for the College (CMO)-USA geomagnetic observatory are presented on the interval 1–31 July 2021; vertical blue and red bars are daily distribution of the BPOL and its STDEV; dotted blue line represents 2 days average values of BPOL time series; red star is Mw8.2 earthquake on 29 July 2021; Mw8.2/43 km (magnitude/depth); vertical red arrow indicates a pre-seismic anomalous signal generated on 22 July. (Obs: SUA means USA).

**Figure 3 entropy-24-00274-f003:**
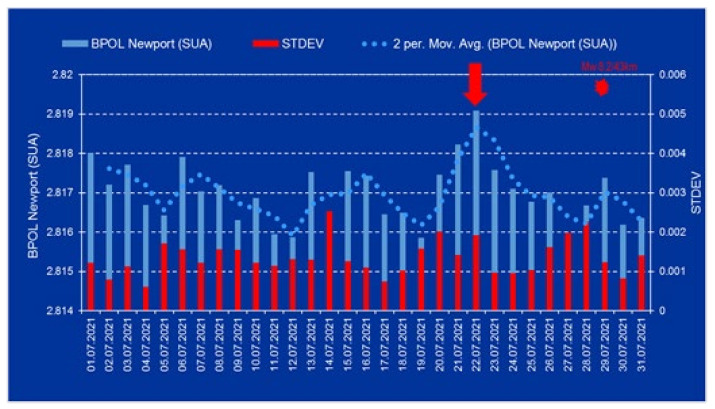
BPOL and STDEV time series for the Newport (NEW)-USA geomagnetic observatory are presented on the interval 1–31 July 2021. Red star is Mw8.2 earthquake on 29 July 2021; Mw8.2/43 km (magnitude/depth); vertical red arrow indicates a pre-seismic anomalous signal generated on 22 July. (Obs: SUA means USA).

**Figure 4 entropy-24-00274-f004:**
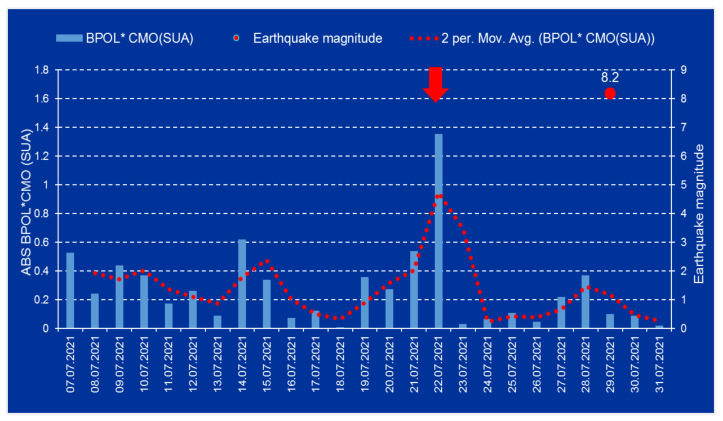
ABS BPOL* time series distribution for College (CMO) observatory, represnted by blue bars, on the interval 7–31 July 2021; red dotted line represents two days average values of the ABS BPOL*; red full circle represents the Mw8.2 eartquake; ABS is BOPL absolute value; vertical red arrow indicates a pre-seismic anomalous signal generated on 22 July. (Obs: SUA means USA).

**Figure 5 entropy-24-00274-f005:**
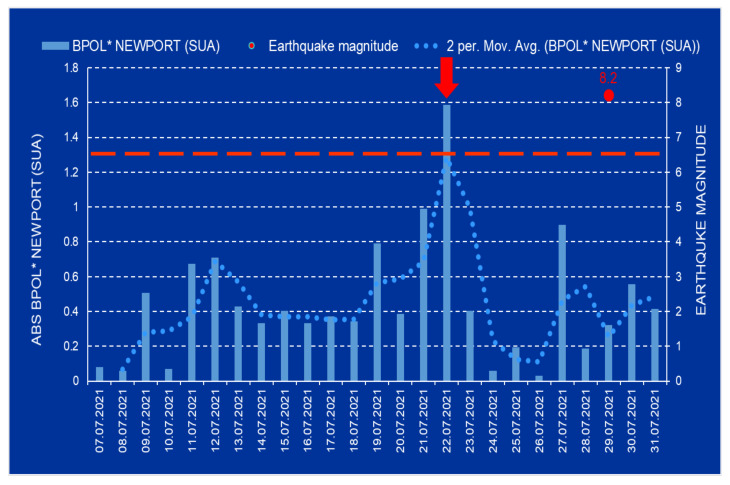
ABS BPOL* time series distribution for Newport (NEW) observatory on the interval 7–31 July 2021. For explanation see Figure 4 caption (Obs: SUA means USA).

**Figure 6 entropy-24-00274-f006:**
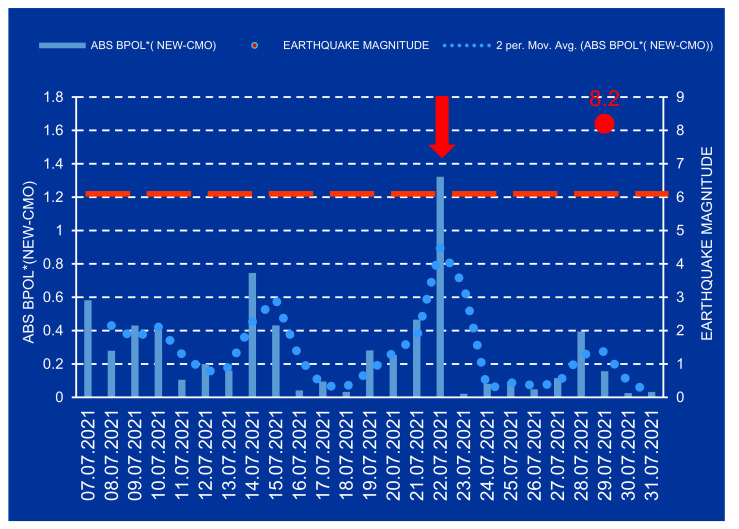
ABS POL*(NEW-CMO) time series on the interval 7–31 July2021; dashed red line represents the threshold for anomaly using STDEV, neccesary to identify the pre-seismic anomalous signal market on the Figure by the red vertical arrow. (Obs: SUA means USA).

## Data Availability

Not applicable.

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
