# Peer review of "Anomalous Geomagnetic Signal Emphasized before the Mw8.2 Coastal Alaska Earthquake, Occurred on 29 July 2021"

_entropy, 2022, doi:10.3390/e24020274_

Round 1

Reviewer 1 Report

This manuscript presents the application of a method already developed and applied to other big earthquakes by the author and his collaborators. In this sense one can say the manuscript does not present any scientific novelty. Even in such case I think the manuscript deserve publication after some corrections and clarifications.

Figure 3 is equal to figure 2. Please, replace it by the correct figure.

The use of equation (2) is not properly explained. Please do that.

Parameters in equations (3) and (4) needs to be explained. The reader can see details in the mentioned references, but the manuscript must be self-consistent.

Author Response

Yes, sorry, my mistake! I changed the figure 3. Thank you!

Equations (2) , (3)and (4): I put new explanations in the text.

Reviewer 2 Report

In this manuscript (ms), the author presents precursory magnetic field variations observed before the 2021 Chignik earthquake of magnitude Mw8.2 that occurred off the coast of the Alaska Peninsula on 29 July 2021 at 06:15:49 UTC. The author analyses freely available data from two geomagnetic stations labeled CMO and NEW of the http://www.intermagnet.org network and identifies anomalous variations mainly on 22 July 2021, i.e, almost one week before the strong earthquake occurrence. This is an important observation that certainly merits publication in entropy. The presentation of the paper, however, needs improvement along the following lines:

1)The author should necessarily update the literature and extend certain portions of Section “1.Introduction” as follows:

Lines 38-39 now read “geomagnetic diurnal variation … Tohoku earthquake [7];” and should be extended as follows: “anomalous geomagnetic diurnal variation mainly in the z-component (which from Maxwell equations reflects the coexistence of seismic electrical activity [J. Geophys. Res. Space Physics 119, 9192-9206 (2014); Europhysics Letters (EPL) 132, 29001(2020)]);”

Lines 43-44 now read: “long-range anomalous … Mw9.0 Great Tohoku earthquake [13];” should be completed as follows: “long-range … Mw9.0 Great Tohoku earthquake [13], the epicenter of which could be estimated [Proc. Natl. Acad. Sci. USA 112, 986-989 (2015)] in advance by studying the spatiotemporal variations of the variability [Europhysics Letters (EPL) 91, 59001 (2010)] of the order parameter of seismicity in Japan;”

2)Figures 1 and 3 should be improved: In Figure 1, the author should check the location of the geomagnetic stations CMO and NEW to be as close as possible to those reported by INTERMAGNET and discuss in the caption what the yellow and brown circles differentiate. Moreover, in the text the author should discuss what the data from the Shumagin station (SHU) show during the period of the present study. Figure 3 is just the same as Figure 2; the data from NEW should appear in Figure 3. Please also fix the typo “SUA” -> “USA” in Figures 2 to 5.

3)The author should provide in a self-contained fashion within this paper all the definitions of the quantities used in Equations (3) and (4). These definitions should replace the text in lines 103 and 109; Reference [20] as well as [Entropy 2019, 21, 29; doi:10.3390/e21010029] where parts of these definitions have been introduced should be also cited for the readers’ better information.

4)All the abbreviations used should be defined in the text, e.g., STDEV, as well as clarify in the captions how the 2 day moving average used is calculated, e.g., avg=(today’s mean value+yesterday’s mean value)/2.0    

5)Refences should be updated and corrected, e.g., in Ref.7 lines 217-218 “431-434” should become “321-326” and the doi link does not work, in Ref.11, line 226, in the doi link “.04.” should become “.05.” for the link to be operable, in Ref.12, the doi link in line 228 is irrelevant, the doi links in lines 232 and 238 do not work and should be fixed. Finally, Ref. 19 in line 240 has the same title, publisher and year as Ref.18.

Finally, a few typing errors like “:.”-> “:” in line 69, “.com” -> “.org” in line 95,  “Refference” -> “Reference” in line 109 , and “1.2” -> “1.3” in line 178.

In summary, I think that the author should revise the ms along the lines mentioned above. I will be glad to suggest publication of an appropriately revised ms.

Author Response

First of all, I want to sincerely thank you for your hard work in helping me to improve the quality of the manuscript. I appreciate a lot !

1) I updated in the manuscript the literature, I inserted your kindly recommendations in lines 38-39 and 43-44, in Figure 1, yellow and brown circles are explained above the figure: yellow: previous 7 days and brown: previous 48 hours;

2)Yes, my mistake, I changed the Figure 3 and of course, SUA becomes USA.

3) I improved the definitions of the relations 3 and 4

4) I put in text the definition for STDEV

5) I made all the changes as you asked, but I don't know how to change the doi because this doi is printed so on the papers what I have.

Round 2

Reviewer 2 Report

The author has successfully improved the manuscript along the lines suggested in my previous review.

There are, however, typing errors that should be corrected before I can suggest publication:

In line 40, reference J. Geophys. Res. Space Physics 119, 9192-9206 (2014) with doi https://doi.org/10.1002/2014JA020580 should be included in the reference list with number 7.

In lines 40-41, reference Europhysics Letters (EPL) 132, 29001(2020) available at

https://iopscience.iop.org/article/10.1209/0295-5075/132/29001 should be included in the reference list with number 8.

In line 41, “[7]” should become “[7-9]” and reference “7.” of the current reference list in line 245 should be renumbered as “9.”

In line 42, “[8]” should become “[10]” and reference “8.” of the current reference list in line 248 should be renumbered as “10.”

In line 43, “[9]” and “[10]” should become “[11]” and “[12]” and references “9.” and “10.” of the current reference list in lines 250 and 252 should be renumbered as “11.” and “12.”

In line 44, “[11]” should become “[13]” and reference “11.” of the current reference list in line 254 should be renumbered as “13.”

In line 45, “[12]” should become “[14]” and reference “12.” of the current reference list in line 256 should be renumbered as “14.”

In line 47, reference Proc. Natl. Acad. Sci. USA 112, 986-989 (2015) available at https://www.pnas.org/content/112/4/986 should be included in the reference list with number 15 and “[15]” should replace “[Proc. Natl. Acad. Sci. USA 112, 986-989 (2015)]” in same line (l.47).

In lines 48-49, reference Europhysics Letters (EPL) 91, 59001 (2010) with doi

 https://doi.org/10.1209/0295-5075/91/59001 should be included in the reference list with number 16 and “[16]” should replace “[Europhysics Letters (EPL) 91, 59001 (2010)]” in line 48.

In line 49, “[13]” should become “[17]” and reference “13.” of the current reference list in line 258 should be renumbered as “17.”

In line 50, “[14]” should become “[18]” and reference “14.” of the current reference list in line 260 should be renumbered as “18.”

In line 51, “[15-20]” should become “[19-24]” and references “15.”, “16.”, “17.”, “18.”, “19.”, and “20.” of the current reference list in lines 262, 263, 265, 268, 270, and 279, should be renumbered as “19.”, “20.”, “21.”, “22.”, “23.” and “24.”, respectively.

In line 82, “[13]” should become “[17]”

In line 84, “[18]” should become “[22]”

In line 92, “[8]” should become “[10]”

In line 115, “[19]” should become “[23]”

In line 128, “[19]” should become “[23]”

As concerns doi’s in the reference list, the following changes are necessary in line 247 “.jseas.” should become “.jseaes.” and in line 257 http://dx.doi.org/10.2183/pjab.86.257 should be replaced by https://doi.org/10.1023/A:1006500408086

In summary, I suggest minor revision in which the author will correct the above typing errors. I will be glad to suggest acceptance of an appropriately revised manuscript.